# From Single Organisms to Communities: Modeling Methanotrophs and Their Satellites

**DOI:** 10.3390/microorganisms14010003

**Published:** 2025-12-19

**Authors:** Maryam A. Esembaeva, Ekaterina V. Melikhova, Vladislav A. Kachnov, Mikhail A. Kulyashov

**Affiliations:** Department of Computational Biology, Scientific Center of Genetics and Life Sciences, Sirius University of Science and Technology, Sirius 354340, Russia; esembaeva.ma@talantiuspeh.ru (M.A.E.); melihova.ev@talantiuspeh.ru (E.V.M.); kachnov.va@talantiuspeh.ru (V.A.K.)

**Keywords:** aerobic methanotrophs, heterotrophic satellites, microbial community, GSM, FBA, community metabolic modeling, metabolic interactions

## Abstract

Aerobic methanotrophs mediate methane oxidation contributing to a major biological sink that limits CH_4_ release to the atmosphere in oxygenated environments and serve as promising platforms for biotechnological applications. In natural and engineered environments, these bacteria rarely exist in isolation but form stable associations with heterotrophic satellites that utilize methanotrophic metabolites, remove inhibitory intermediates, and provide essential growth factors. Such interactions enhance methane oxidation efficiency and community stability, yet the metabolic mechanisms underlying them remain poorly resolved. This review summarizes current knowledge on both natural and synthetic aerobic methanotrophic consortia, focusing on the composition, functions, and biotechnological relevance of satellite microorganisms. We systematically examine available mathematical frameworks—from ecological and statistical models to genome-scale metabolic reconstructions and dynamic flux balance analysis—applied to methanotrophs and their satellites. Our analysis reveals that while genome-scale metabolic models have been developed for model heterotrophic species, only a few correspond to experimentally identified methanotroph satellites, and community-level reconstructions remain limited. The lack of curated and experimentally validated models restricts the predictive power of current approaches. Addressing these limitations will require not only targeted reconstruction of satellite metabolism, but also the combined use of complementary computational methods followed by experimental verification. Such an integrative strategy will be essential for understanding methanotrophic community organization and function and, more broadly, other microbial consortia with complex metabolic interactions. Addressing these limitations through targeted reconstruction of satellite metabolism and integration of existing models will be key to advancing quantitative understanding of methanotrophic community organization and function.

## 1. Introduction

Aerobic methane oxidation is an important biological sink that limits CH_4_ fluxes to the atmosphere in oxic and microoxic environments. Aerobic methanotrophic bacteria (methanotrophs) play a key role in this process and their ability to oxidize methane is determined by the presence of methane monooxygenase, the enzyme that catalyzes the first step of methane oxidation to methanol. Methane monooxygenase (MMO) occurs in two distinct isoforms: particulate methane monooxygenase (pMMO), embedded in cell membranes, and soluble methane monooxygenase (sMMO), dissolved in the cytoplasm [1,2,3,4], with different methanotroph strains encoding either one or both of these isoforms. In the latter case, the expression of the corresponding isoform is regulated by the copper concentration in the medium: at high copper content, pMMO is predominantly expressed. In addition to methane, this enzyme is capable of oxidizing other hydrocarbons, in particular, components of natural gas such as ethane and propane, which can lead to the accumulation of by-products in the form of organic acids that inhibit the growth of methanotrophs [1,2,3,5]. Accumulated data indicate that stable mixed cultures, including both methanotrophic and heterotrophic microorganisms (where heterotrophs refer to non-methanotrophic members that coexist with methanotrophs and utilize their metabolic by-products), are formed during long-term continuous cultivation of methanotrophs on natural gas under non-sterile conditions [6,7]. Heterotrophic satellites in such consortia perform a variety of functions, including supplying methanotrophs with growth factors [7,8], as well as increasing their adaptive abilities under conditions of nutrient deficiency and other limiting factors [9]. However, one of the main known functions of these satellite microorganisms is the stabilization of methanotroph growth through utilization of metabolic by-products such as methanol, formaldehyde, and organic acids (acetate, lactate, malate) as carbon and energy sources [6,7,10,11,12]. Thus, the participation of satellites can be a critical factor in long-term cultivation of methanotrophs under natural and biotechnological conditions. However, the structure of methanotrophic communities, as well as the mechanisms of interactions between the main bacterial strain and its satellites, remains insufficiently understood, primarily due to methodological and analytical constraints [13].

Traditional methods for analyzing microbial communities typically rely on isolation, cultivation, and subsequent characterization of individual microorganisms. Despite technological advances, this approach remains limited by the inability to cultivate a substantial fraction of microorganisms outside their native community context [14,15,16]. At the same time, modern molecular biology methods for determining the composition of microbial communities make it possible to bypass the need for laboratory cultivation. These culture-independent methods include polymerase chain reaction (PCR) based approaches targeting conserved genomic regions; techniques exploiting sequence or sequence-length polymorphisms, such as DGGE (denaturating gradient gel electrophoresis)/TTGE (temporal temperature gradient gel electrophoresis), SSCP (single-strand conformation polymorphism), RAPD (random amplified polymorphic DNA), ARDRA (amplified rDNA restriction analysis), T-RFLP (terminal restriction fragment length polymorphism), LH-PCR (length heterogeneity PCR), and RISA (ribosomal intergenic spacer analysis); real-time PCR (RT-PCR); fluorescence in situ hybridization (FISH); fatty acid methyl ester (FAME) analysis; as well as 16S or whole-genome metagenomic sequencing methods [15,16,17]. However, molecular methods still require an integrated approach that would encompass different methods for studying the composition of the microbial community and the quantitative representation of organisms [15,16,17]. In addition to characterizing microbial community composition, elucidating interactions within these communities represents a significant analytical objective. To address this, RNA-based techniques are frequently employed: ribosomal RNA (rRNA) analysis enables inference of microbial growth dynamics, while messenger RNA (mRNA) profiling provides insights into gene expression patterns occurring during microbial co-cultivation. Furthermore, stable isotope-based methods, including stable isotope probing (SIP), MAR-FISH, and Raman-FISH, facilitate detailed assessment of metabolic activities at single-cell resolution. However, these techniques are currently limited by constraints on throughput scale, the need for advanced instrumentation, and complexities associated with interpreting the resulting datasets [16,18].

Thus, even taking into account the advantages of mixed methanotrophic cultures, their use at the current stage resembles interaction with a “black box”. To unlock their full potential, it is necessary to move beyond descriptive studies toward predictive frameworks. Systems biology offers the necessary tools to decipher complex interspecies interactions. However, a systematic link between satellite taxonomy, metabolic roles, and available modeling resources is currently missing. This review compiles satellite organisms reported in both natural and synthetic methanotroph-associated communities and categorizes them according to their effects on community function and their potential relevance to bioremediation and biotechnology. The taxonomy-function overview is linked to systems-biology approaches that can provide a mechanistic understanding of key features of such consortia. We also evaluate the current landscape of genome-scale metabolic (GSM) modeling for methanotroph-associated organisms. A critical shortage of strain-specific GSMs for satellite species and community-level GSM reconstructions for methanotroph–heterotroph consortia are highlighted, as these limitations complicate the rational design of synthetic methanotrophic communities.

## 2. System Biology Approaches for Analysis of Microbial Metabolic Networks and Community Interactions

The shift from monoculture-based processes to community-level strategies reflects growing recognition that microbial consortia can be more robust and metabolically efficient than individual strains [19]. Understanding the functioning of natural communities has contributed to the growing interest in designing synthetic communities to solve various biotechnological problems. However, successful implementations of microbial community applications remain limited, which is explained by the limited understanding of the quantitative composition of organisms and interspecies interactions [20]. Current descriptions of microbial communities are often primarily descriptive: they indicate which taxa are present, how their abundances change and which groups tend to co-occur, but they provide limited guidance on how deliberate perturbations or design choices will affect system-level behavior. For biotechnological applications, such as bioremediation or methane-based production processes, there is therefore a clear need for mechanistic models that explicitly link the physiology and metabolism of individual members to emergent community functions. The expanding role of microorganisms in biotechnology and bioremediation has driven the advancement of systems biology approaches, enabling the use of mathematical models to predict community dynamics and dissect underlying microbial interactions [21]. Several recent reviews [21,22,23,24,25,26] provide detailed overviews of currently available approaches, including their capabilities, limitations and practical applications. Therefore, in this section we only briefly summarize the main applications of these approaches. We also highlight which of them can provide a detailed mechanistic understanding of community function and which have so far been applied to methanotroph–heterotroph communities (Table 1).

A wide range of computational strategies was proposed to analyze microbial communities and can be divided into two main groups: data-driven methods based on statistics approaches and mechanistic mathematical models, including ecological and cell-level models (Table 1).

Data-driven methods have computational flexibility, identify structures and patterns in large amounts of data, but do not provide a mechanistic description of processes at the cell level [24]. For example, co-occurrence network analysis makes it possible to identify potential interactions between taxa, identify key species, and formulate hypotheses about the structure of the community [24,27,28]. The time-series correlation method was used to study the dynamics of a community, identify a sequence of events, and determine transitional states. Causal inference models were used to identify directional interactions between species and allow testing hypotheses about the influence of certain taxa on community dynamics [24]. Regression models were employed to predict dynamics based on previous system states [22,27]. In the context of methanotrophic communities, these methods can be used as a first step to generate hypotheses about which heterotrophs may act as stable satellites or key cross-feeding partners [29]. At the same time, these approaches mainly indicate which interactions are likely to exist, without revealing the metabolic pathways or regulatory mechanisms that give rise to them [24].

In contrast to statistical methods, mathematical models seek to formalize the mechanisms underlying the dynamics of microbial communities. Ecological models describe the behavior of populations and their interactions [22]. Classical Lotka–Volterra systems were used to model competitive and cooperative relationships between species, remaining one of the most common models due to its simplicity and interpretability, although they have limited ability to account for complex metabolic interactions [30,31]. Resource-explicit models, like the MacArthur model, serve as the basis for analyzing competition for resources, whereas Monod-type kinetics describes the dependence of growth on substrate concentration, which allows them to be used to predict community resilience and response to environmental changes [21,22,32]. Together, such models were used to predict how communities respond to shifts in resource supply, disturbance regimes or invasion by new taxa. In the context of methanotrophic systems, such ecological models can be applied to study competition between methanotrophs and other aerobic or facultatively anaerobic taxa for oxygen and carbon sources [33].

Another important class of mechanistic approaches is agent-based modeling, which represents individual cells in an explicit spatial environment. These models are typically used to investigate phenotypic heterogeneity, biofilm formation, spatial segregation of species and the impact of local nutrient or metabolite gradients on community dynamics [21,26]. For methanotroph-based consortia, agent-based simulations are particularly useful for exploring how microscale oxygen and methane gradients shape the spatial organization of methanotrophs and their heterotrophic satellites and how this organization feeds back on methane oxidation efficiency. Organism-level models provide an even more detailed description of interaction mechanisms by explicitly accounting for cellular metabolism. Genome-scale metabolic models (GSM) make it possible to reconstruct the metabolic networks of organisms, determine metabolic dependencies, predict cross-feeding interactions and quantify the contribution of each taxon to community-level functions. Community FBA combines several GSMs into a multi-species system and can be used to infer potential mutualistic, competitive or parasitic relationships based on simulated nutrient flows and metabolite exchange. Its dynamic extension, dFBA is used to investigate how community composition and fluxes change over time under fluctuating substrate availability [24]. In the context of methanotroph-satellite communities, GSMs are particularly valuable for predicting how carbon and reducing equivalents derived from methane are partitioned among community members and for guiding the rational design of synthetic co-cultures [34,35,36]. Using a single approach to modeling microbial communities can limit the completeness of the results obtained; therefore, integrative strategies provide the greatest value [22,37]. For example, the combination of GSM modeling with network analysis reveals metabolic interactions and active functional modules, and the integration of GSM with dynamic FBA or agent-based approach allows for taking into account the spatial organization, distribution of species and diffusion of metabolites, creating a more realistic representation of the structure and dynamics of communities.

Overall, the approaches discussed address different questions that arise when modelling microbial communities. Data-driven methods are mainly used to identify taxa that are consistently associated with methanotrophs and to detect potential cross-feeding pairs. Ecological models are suited to questions about substrate competition within consortia and about how methanotroph–heterotroph communities respond to changes in environmental conditions. GSMs and their dynamic extensions, as well as integrative approaches, enable quantitative analysis of metabolic cross-feeding and of carbon flux distribution within the community, and they support the rational design of synthetic consortia.

As summarized in Table 1, most statistical and ecological approaches do not provide a mechanistic description at the organism level and have only rarely been applied to methanotroph–heterotroph communities. In contrast, GSM-based models, particularly when combined with other mechanistic frameworks such as dynamic FBA, offer both organism-level resolution and direct applicability to such systems. For this reason, in the following chapters we focus on aerobic methanotrophs, their satellite organisms and the available GSM reconstructions that can be used to build community-level models of methanotroph–heterotroph consortia.

## 3. Methanotroph’s Satellites

Methanotrophs are the primary drivers of methanotrophic communities, utilizing methane as a carbon and energy source and generating metabolites that support the growth of coexisting heterotrophs (Figure 1). Among these metabolites are organic acids such as acetate, formate, succinate, and lactate [26], which can inhibit methanotroph growth, as well as methanol [27], formaldehyde [28], exopolysaccharides [29], and products released through cell lysis [6,30,31]. A central hypothesis regarding the role of heterotrophic satellite organisms is that they facilitate the removal of inhibitory metabolites, thereby alleviating metabolic stress and promoting increased growth rates of methanotrophs [32]. In addition, satellites can stimulate methanotroph growth by producing growth factors, which has been shown when cultivating some strains of methanotrophs with representatives of *Rhizobiaceae* [32].

### 3.1. Natural Methanotroph’s Satellites

Despite the critical role of companion heterotrophs in methanotrophic communities, their identification and the precise elucidation of interaction mechanisms remain challenging (see Section 1). Nevertheless, for several methanotrophic strains, available data highlight specific associated heterotrophs and suggest potential interaction mechanisms observed during co-cultivation (Table 2).

#### 3.1.1. Satellites That Promote a Methanotroph’s Growth by Consuming Metabolic By-Products

Common satellites of methanotrophs include representatives of the genera *Cupriavidus*, *Brevibacillus*, *Bacillus*, and *Aneurinibacillus* [6,7,10,11,12]. These organisms maintain stable methanotroph growth by removing metabolic by-products and lysed-cell debris [6,40,45]. Among the most frequently observed companion organisms in methanotrophic communities, Gram-negative bacteria of the genus *Cupriavidus* are of particular interest. These bacteria are well known for their roles in bioremediation, including the removal of CO_2_ and various toxic compounds [46,47,48], as well as in biotechnological applications such as the production of polyhydroxyalkanoates (PHA) [47,49]. Gram-positive companions identified in these communities include representatives of the genera *Aneurinibacillus*, *Bacillus*, and *Brevibacillus*, with *Bacillus cereus* and *Brevibacillus agri* among the notable species [6]. A less frequently observed group of associated organisms includes representatives of the genus *Acidovorax*, capable of consuming acetate produced by methanotrophs [50].

In addition to these heterotrophic satellites, methanotrophic communities commonly include non-methanotrophic methylotrophs of the family *Methylophilaceae*. Members of this family form close associations with methanotrophs of the *Methylococcaceae* and frequently dominate satellite communities. This dominance reflects their ability to outcompete other methanol-utilizing organisms for methanol released by methanotrophs [9,42,51]. The most frequently encountered *Methylophilaceae* satellites belong to the genus *Methylotenera* [43,50,52]. The composition of these satellite communities was reported to shift depending on oxygen availability under controlled laboratory conditions: *Methylotenera* dominated under low-oxygen conditions where *Methylobacter* were the primary methanotrophs. Increasing oxygen availability promotes higher abundances of *Methylosarcina* alongside *Methylobacter*, which leads to a replacement of *Methylotenera* by *Methylophilus* [50].

Beyond *Methylophilaceae*, permanent satellites of methanotrophs also include representatives of the genera *Flavobacterium*, *Chryseobacterium*, *Pseudomonas*, and *Sphingopyxis*. These heterotrophs remain active under oxygen- and nitrogen-limited conditions [41]. A study by [41] demonstrated that in controlled laboratory experiments *Chryseobacterium* species often dominate these communities, although under a methane-to-oxygen ratio of 1:9, *Flavobacterium* representatives become more abundant. This suggests that oxygen availability significantly shapes the composition of heterotrophic satellites in methanotrophic consortia. Competitive interactions between *Chryseobacterium* and *Flavobacterium* for similar ecological niches have been proposed, with *Chryseobacterium* generally displacing *Flavobacterium* in community settings [41]. Notably, *Chryseobacterium* maintains high abundance even under elevated methane concentrations, when other heterotrophic companions are suppressed by reduced oxygen levels [41,53]. Beyond their role in methanotrophic consortia, these bacteria are known for promoting plant growth through hormonal stimulation and for their antagonistic effects on plant pathogens [54,55]. They also hold bioremediation potential, being capable of degrading various pollutants, including polycyclic aromatic hydrocarbons [56,57,58], pentachlorophenol, aniline, and carbofuran [41,58]. Interactions between *Methylosarcina* and *Chryseobacterium* may further enhance the efficiency of bioremediation processes within methanotrophic communities [41].

Finally, methanotrophic consortia often include Gram-negative bacteria of the genera *Pseudoxanthomonas* and *Piscinibacter*, as well as representatives of the family *Rhodocyclaceae*, which are capable of reducing selenate [59]. These organisms play an important role in removing inhibitory metabolites such as acetate, lactate, formate, propionate and butyrate [59], and are widely used in bioremediation [60,61,62].

#### 3.1.2. Satellites That Promote Methanotroph Growth by Producing Growth Factors

Among the natural satellites of methanotrophs, members of the family *Rhizobiaceae,* particularly *Rhizobium* sp., *Mesorhizobium* sp., and *Sinorhizobium* sp., are particularly noteworthy due to their role in producing essential growth factors such as cobalamin (vitamin B_12_) [8]. The dependence of certain methanotrophs on *Rhizobiaceae*-derived factors is evident from their limited or absent growth in cobalamin-free media, which likely to underlie the formation of stable associations between these partners. For example, representatives of the genera *Methylovulum*, *Methyloparacoccus*, and *Methylomonas* establish persistent interactions with *Rhizobiaceae* through the exchange of cobalamin [38,63]. This is further supported by experimental data: co-cultivation of *Rhizobium* sp. 122 with *Methylovulum miyakonense* HT12 stimulated methanotroph growth via cobalamin production [8], whereas the same *Rhizobium* sp. 122 strain enhanced the growth of *Methylobacter luteus* through additional, non-cobalamin growth factors [8].

### 3.2. Potential Methanotroph’s Satellites

In addition to natural methanotrophic communities, methanotrophs can also be grown in synthetic consortia, where satellite species are chosen to meet specific aims, such as increasing growth rates, improving methane assimilation or boosting the production of selected metabolites (Figure 1). Table 3 summarizes all synthetic methanotroph–heterotroph consortia that have been experimentally characterized to date.

#### 3.2.1. Synthetic Methanotrophic Communities for the Production of Metabolites

Potential satellites of methanotrophs include the Gram-negative bacterium *Pseudomonas putida* KT2440, a well-characterized strain widely employed in the bioremediation of soils contaminated with organic pollutants, particularly aromatic compounds [73,74]. Unlike many other members of the species, *Pseudomonas putida* KT2440 is classified as a non-pathogenic strain (biosafety level 1) making it particularly well suited for biotechnological applications and one of the most preferred candidates for use in engineered microbial consortia [73] for polyhydroxyalkanoates (PHA) production [75] and pharmaceuticals [76]. Another potential satellite is *Rhodococcus opacus* DSM 43205, a facultative lithoautotrophic Gram-positive bacterium of interest due to its ability to convert CO_2_ into value-added products [77,78,79,80]. In particular, the potential for using *R. opacus* DSM 43205 for the production of beta-alanine and lactate when grown on CO_2_ as the sole carbon source has been demonstrated [77]. In a recent study [68], *Pseudomonas putida* KT2440 and *Rhodococcus opacus* DSM 43205 were employed in synthetic consortia with *Methylocystis parvus* OBBP and *Methylocystis hirsuta* CSC1—both type II methanotrophs capable of accumulating PHA when cultivated on methane. Challenges related to mass transfer in bioreactor systems often limit biopolymer synthesis due to the accumulation of inhibitory metabolites such as formaldehyde and formate. The inclusion of *Pseudomonas* and *Rhodococcus* species, which utilize these by-products and synthesize PHA themselves, enhances overall process performance and metabolic efficiency in methanotrophic systems [68].

In line with this, a recent study evaluated four synthetic community combinations of methanotrophs and heterotrophs: *M. hirsuta* with *R. opacus*; *M. hirsuta*, *R. opacus*, and *P. putida*; *M. parvus* with *R. opacus*; and *M. parvus*, *R. opacus*, and *P. putida*, alongside monoculture controls [81]. All configurations, except the *M. parvus*–*R. opacus*–*P. putida* consortium, demonstrated improved substrate consumption and enhanced PHA accumulation, with the highest yields achieved by the *M. hirsuta*–*R. opacus* pairing. The reduced efficiency observed in the *M. parvus*–*R. opacus*–*P. putida* community is likely due to delayed substrate uptake by the methanotroph, potentially caused by *P. putida* competing for oxygen during co-cultivation, which negatively impacted methane assimilation by *M. parvus*. The authors noted that the inhibitory effect observed in certain community configurations could be mitigated by optimizing the ratio of organisms within the consortium. In a follow-up experiment, the potential of a synthetic of *M. parvus* and *P. putida* community for polymer synthesis was investigated using a mixture of valerate propionate, acetate, and butyrate as co-substrates instead of valeric acid. Consistent with prior observations, co-cultivation enhanced methane assimilation and increased poly(3-hydroxybutyrate-co-3-hydroxyvalerate) (PHBV) yields compared to monocultures [81].

Another potential companion is *Cupriavidus necator* LMG 1201. In the study by Kerckhof et al. 2021 [82], an experiment was conducted to evaluate co-cultivation strategies of methanotrophs and heterotrophs for the production of microbial protein with an optimized amino acid composition. Among the tested synthetic consortia, the combination of *Methyloparacoccus murrelli* LMG 27482 and *Cupriavidus necator* LMG 1201 demonstrated the highest protein yield along with a favorable amino acid profile, making it the most promising pairing for single-cell protein production [82]. Also, companions of the genus *Cupriavidus* include *C. taiwanensis* LMG19424, a diazotrophic bacterium, isolated from mimosa tubers [83,84]. This strain shows similarity to other representatives of *Cupriavidus*, such as *C. necator* H16 [85], but due to the pRalta plasmid, it is able to form symbiotic relationships [86]. In the work [87], the stimulatory effect on *Arabidopsis thaliana* was investigated upon inoculation with *C. taiwanensis* LMG19424, due to the strain’s ability to fix nitrogen. Co-cultivation of *C. taiwanensis* LMG19424 and *Methylomonas* sp. M5 stimulated methanotroph growth [38]; though the underlying mechanisms remain unclear. Similarly, *Staphylococcus aureus* R-23700 enhanced *Methylosarcina fibrata* DSM 13736 growth. However, the pathogenicity of the *S. aureus* species and the lack of mechanistic data limit its practical application in synthetic consortia [40].

*Escherichia coli* represents a well-established biotechnology platform for value-added chemical synthesis in methanotrophic consortia [88]. For example, *E. coli* strain SBA01 was co-cultivated with *Methylococcus capsulatus* Bath on methane as the sole carbon source [65]. Under these conditions, *E. coli* utilized organic acids (acetate, malate, and succinate) released by the methanotroph to synthesize mevalonate [65]. In another study, *E. coli* BL21 (DE3), engineered for polyhydroxybutyrate (PHB) production, was co-cultured with *Methylocystis* sp. OK1 [70]. In this system, propane was oxidized by *Methylocystis* sp. OK1 to acetone, which served as a substrate for enhanced PHB synthesis by *E. coli.* Additionally, other organic acids generated by the methanotroph were used by *E. coli* as carbon sources [70].

#### 3.2.2. Synthetic Methanotrophic Communities for Bioremediation

In the study by [66], a synthetic consortium was developed for efficient atmospheric nitrogen fixation, comprising the methanotroph *Methylotuvimicrobium buryatense* 5GB1, engineered with the plasmid pAMR4-dtom1, and the diazotroph *Azotobacter vinelandii* M5I3. Nitrogen fixation is catalyzed by nitrogenase, an oxygen-sensitive enzyme that typically operates under anaerobic conditions. Although *A. vinelandii* is capable of fixing nitrogen under aerobic conditions, the process of breaking the triple bond in the N_2_ molecule is energetically demanding, requiring substantial ATP and reducing equivalents. These energy demands can be met through the oxidation of lactate, highlighting the importance of metabolic cooperation within the consortium [89]. At the same time, *M. buryatense* carries the pAMR4-dtom1 plasmid with the lactate dehydrogenase gene (*Lhldh*) from *Lactobacillus helveticus* for the formation of L-lactate from pyruvate [90]. Under conditions where CH_4_ and N_2_ served as the sole carbon and nitrogen sources, respectively, the engineered methanotroph produced lactate, which was subsequently utilized by *A. vinelandii* to support its carbon metabolism and energy generation. In turn, *M. buryatense* consumed part of the ammonium reduced by *A. vinelandii* from atmospheric nitrogen [66].

Cyanobacteria *Arthrospira platensis* NIES-39 [67] and *Synechococcus* PCC 7002 [69] can also be used as synthetic satellites of methanotrophs. Cultivation with them can contribute to the efficient removal of CH_4_ and CO_2_ from the environment, as well as overcome the problem of mass transfer when growing methanotrophs in bioreactors, due to additional O_2_ production. *A. platensis* is the most cultivated bacterium capable of photosynthetic activity and is used in the food and pharmaceutical industries due to its rich amino acid composition and probiotic properties [91]. Co-cultivation with *Methylotuvimicrobium buryatense* 5GB1 resulted in increased growth rates of both organisms compared to their monoculture. The community was found to exchange oxygen and carbon dioxide in the environment; however, it was suggested that there are more complex internal interactions that favorably affect the growth of both strains [67]. Co-cultivation of *Synechococcus* PCC 7002, which has potential for use in the biotechnology industry [92], with *Methylotuvimicrobium alcaliphilum* 20Z also showed the dependence of the methanotroph under anaerobic environmental conditions on oxygen produced by the cyanobacterium [69].

It was also shown [93] that one of the potential satellites of methanotrophs is the perchlorate-reducing bacterium *Dechloromonas agitata*, which is capable of reducing perchlorate to chloride under anaerobic conditions with the concomitant release of oxygen [94,95], providing a potential oxygen source for methanotrophs. Co-cultivation of *D. agitata* with *Methylococcus capsulatus* demonstrated that *D. agitata* utilized acetate secreted by the methanotroph as a carbon source, facilitating perchlorate reduction and simultaneous oxygen production needed by *M. capsulatus* [64,93]. Furthermore, this cross-feeding interaction appears to be a general strategy, as evidenced by the co-culture of another aerobic methanotroph, *Methylomonas* sp. LW13, with *Dechloromonas agitata* CKB [64]. Such mutualistic interactions may be especially valuable in ecosystems affected by perchlorate contamination.

In the study [71], *Methylomonas methanica* NCIMB 11130^T^ was shown to exert a stimulating effect on methanotrophs when co-cultivated with *Rhizobium radiobacter* [71], *Ochrobactrum anthropi*, *Pseudomonas putida*, and *Escherichia coli*. Among these companion organisms, *O. anthropi* was not characterized in detail in this context, and it is important to note its pathogenic potential [96]. However, the study [71] did not explore the underlying interactions within the communities that could explain the observed stimulatory effects.

## 4. Mathematical Models of Methanotroph’s Satellites

Based on the above, satellite species play an important role both in natural and synthetic methanotrophic communities. They can improve methane assimilation and methanotroph growth and facilitate applications ranging from bioremediation to the biosynthesis of valuable metabolites. Despite the community-level approaches described earlier (Section 2), a deeper understanding of how stable interactions form in these systems requires organism-specific models. In this context, GSM modeling plays a key role, as it allows for the consideration of the metabolic characteristics of each community member. However, we firstly need to create new or find available models of every microorganism, which will be used as a component of the community model reconstruction.

Despite a large number of individual GSM models reconstructed in recent years [97,98], there is still a lack of models for natural satellites associated with methanotrophs. This chapter provides an overview of published GSM models for methanotrophic satellites, including model with the application of dFBA. Existing models usually cover only closely related strains, mainly because these organisms are difficult to isolate in pure culture and have seen limited use in biotechnology [6,7,10,11,12]. In contrast, synthetic methanotrophic communities tend to incorporate well-characterized and biotechnologically relevant strains. As a result, a range of mathematical models was reconstructed for these organisms to explore their individual metabolic capabilities, providing a foundation for their integration into community-level models.

### 4.1. Genomes-Scale Mathematical Models

Due to the specific advantages of GSM models—which require comparatively less experimental data for reconstruction than many other model types—these models represent the majority of those available for methanotroph satellite organisms. A detailed overview of the GSM models developed for these organisms is provided in Table 4. Given the limited number of strain-specific GSM reconstructions for satellites, we organized the models in this subsection into three categories. Strain-to-strain models compile GSMs reconstructed directly for satellite strains associated with methanotrophs. Models of closely related strains include GSMs developed for taxonomically proximate organisms that, although not isolated as satellites themselves, are expected to exhibit broadly similar metabolic behavior. While the use of such models is not always fully justified, we assume that closely related strains may share key metabolic traits. These models can be further refined and expanded to account for strain-level differences, ultimately enabling reconstruction of a more accurate satellite-specific model. Finally, non-curated models summarize large-scale, automated reconstruction projects that incidentally include certain satellite organisms. The use of these models may be considerably hindered by the lack of manual curation, meaning that additional refinement—such as correction of incomplete pathways or unrealistic growth phenotypes—may be required before they can be reliably integrated into community-level reconstructions.

#### 4.1.1. Strain-to-Strain Models

Some of the satellite organisms for which GSM models were reconstructed are cyanobacteria, including *Arthrospira platensis* NIES-39 [99] and *Synechococcus* sp. PCC 7002 [119]. The model for *A. platensis* NIES-39 was developed by the group [99] and validated against experimental data on growth rates under different substrate conditions. This model was used to explore genetic modifications aimed at optimizing glycogen and ethanol production. Notably, the model is available in XLSX format rather than the standard SBML, which may limit its direct integration into some modeling platforms [99]. In contrast, four GSM models were built for *Synechococcus* sp. PCC 7002, as summarized in the review [121]. These include *i*Syp611 [117], *i*Syp708 [118], *i*Syp728 [119] and *i*Syp821 [120]. All four models were experimentally validated, with *i*Syp611 being the first to be reconstructed. Subsequent models represent expansions of the original to meet specific biotechnological objectives, such as improving photosynthetic efficiency or enhancing the production of biofuels and other valuable compounds [121].

Next, one potential satellite which has a several number of GSM models is *Escherichia coli* BL21(DE3), a widely used host in biotechnology. The first of these was reconstructed in the study [105], which presented genome-scale metabolic (GSM) models for 55 *E. coli* strains. Among them, two models—*i*B21_1397 and *i*ECD_1391—were specifically constructed for *E. coli* BL21(DE3), based on different genome assemblies available in the NCBI database. These models enabled comparative analysis of the metabolic capabilities across *E. coli* strains and helped identify unique metabolic pathways activated under various environmental conditions. All models in the study were generated using automated reconstruction pipelines followed by manual curation to ensure accuracy. Building upon this work, a refined model for *E. coli* BL21(DE3), *i*EC1356_Bl21DE3 [100], was developed to investigate strain-specific metabolic features across seven *E. coli* strains. This model further enhanced the resolution of metabolic comparisons and contributed to the growing toolkit for rational strain engineering in synthetic microbial communities.

GSM models were also constructed for *Pseudomonas putida* KT2440, a strain recognized as a promising platform in biotechnology for the production of a wide variety of metabolites [122]. Its extensive use led to the development of six models, all summarized in the same study [122]. The initial models—*i*JN746 [107], *i*JP815 [108], and PpuMBEL1071 [110]—laid the groundwork for capturing core metabolic functions. The fourth model, *i*JP962 [109], introduced improvements by incorporating comparative genomic analysis with closely related species to enhance metabolic network connectivity [109]. Subsequently, the PpuQY1140 [111] model was developed based on consensus metabolic pathways derived from the previous four models (*i*JN746, *i*JP815, PpuMBEL1071, and *i*JP962) [111]. The most recent and comprehensive reconstruction, *i*JN1462 [112], integrates extensive metabolic features and offers improved accuracy and predictive capabilities. Importantly, all of these models underwent manual curation and were validated against experimental data on growth performance across a range of substrates, ensuring their reliability for further biotechnological and systems biology applications.

#### 4.1.2. Models of Closely Related Strains

In Section 3, *Azotobacter vinelandii* M5I3 strain was described as a methanotroph satellite, but there are currently no strain-specific GSM models available for this organism. However, two GSM models were developed for the closely related *A. vinelandii* DJ strain. The first, *i*DT1278 [100], was constructed to investigate nitrogen metabolism. This model was manually curated and validated using experimental growth data on various carbon and nitrogen sources, ensuring its accuracy and applicability [100]. An extended version of this model, *i*AA1300 [101], includes additional reactions that were missed from *i*DT1278 and incorporates corrections to reaction directionality within central metabolic pathways. *i*AA1300 was used to study the impact of metal availability on nitrogenase activity and culture growth, as well as the organism’s metabolic adaptation to anaerobic conditions in the presence of different nitrogen sources [101]. These models offer valuable insights into nitrogen fixation and energy metabolism in *A. vinelandii*, with potential relevance to synthetic community design.

A similar situation is observed for the satellite strains *Cupriavidus necator* LMG and *Cupriavidus taiwanensis* LMG19424: there are currently no published GSM models available for either organism. However, three models were build for the closely related and biotechnologically important strain *C. necator* H16, as summarized in the recent review [123]. These include RehMBEL1391 [102], a modified version of RehMBEL1391 [103], and *i*CN1361 [104]. The modified RehMBEL1391 model expanded on the original by enhancing metabolite and reaction annotations and was integrated with proteomics data for improved validation [103]. The most recent model, *i*CN1361, was validated using experimental data on growth rates across various carbon sources and transcriptomic analyses under nitrogen-limiting conditions, providing a robust framework for studying metabolic adaptation and resource allocation [104]. Together, these models support the use of *C. necator* H16 in metabolic engineering and synthetic community design, particularly for applications involving carbon and nitrogen flux optimization.

No genome-scale metabolic model is currently available for the satellite strain *Rhodococcus opacus* DSM 43205. Instead, a genome-scale metabolic model *i*GR1773 has been reconstructed for the related strain *R. opacus* PD630, which is known for its potential in triacylglyceride production [113]. The *i*GR1773 model was validated against experimental growth data on various carbon sources, and its predictive accuracy was further enhanced through the integration of transcriptomic data to refine key metabolic features [113]. This model provides a valuable basis for understanding the metabolic potential of *R. opacus* strains and supports their future application in synthetic consortia.

Only models for the closely related strains of *Staphylococcus aureus* R-23700 are currently available, as compiled in the article [124]. These include models for *S. aureus* N315: *i*SB619 [114] and *i*MH551 [115], as well as for *S. aureus* USA300 strain JE2: *i*YS854 [116]. The *i*SB619 model was the first metabolic model constructed for *S. aureus*, though it employed the biomass equation of *Bacillus subtilis*. Later, the *i*MH551 model introduced a species-specific biomass formulation for *S. aureus*, but lacked gene associations for the included reactions, which limits its functional depth [124]. The *i*YS854 model was developed for the USA300 strain based on an automated reconstruction [125] that was subsequently manually refined for improved biological accuracy and completeness [116].

#### 4.1.3. Non-Curated Models

In addition to manually curated and experimentally validated models, several automatically reconstructed models of methanotroph satellites were generated as part of large-scale microbial metabolism initiatives. One such project is Path2Models [126], which produced three types of models: kinetic, logic-based, and constraint-based (GSM), using data from major pathway databases including KEGG [127], MetaCyc [128], and SABIO-RK [129]. As a result of this effort, more than 2600 organism-specific models are currently available through the BioModels database [126] (http://www.ebi.ac.uk/biomodels-main/path2models (accessed on 13 December 2025)).

Another major project is EMBL GEMs, accessible via https://github.com/cdanielmachado/embl_gems (accessed on 13 December 2025). This initiative reconstructed 5587 genome-scale metabolic models for diverse bacterial species using genomic data from the NCBI RefSeq database. The models were generated using the CarveMe (version 1.0.5) automated reconstruction pipeline [130], providing a broad and standardized platform for the exploration of microbial metabolism at scale. These resources greatly expand the availability of draft metabolic models for lesser-studied organisms, including potential methanotroph satellites.

### 4.2. Genome-Scale Mathematical Models with dFBA

Although a significant number of GSM models were developed for methanotroph-associated organisms, only two of them have been extended to support dynamic Flux Balance Analysis (dFBA). In a study by Dodia et al. [131], the dFBA approach was applied to model the metabolic flux dynamics of *Escherichia coli* BL21 (DE3) during fed-batch cultivation with recombinant protein production. The researchers performed a comprehensive analysis of spent media, quantifying 246 extracellular metabolites, which were subsequently incorporated as constraints in the dFBA simulations using an existing genome-scale metabolic model.

A dFBA-capable model was developed for a closely related satellite strain of *Cupriavidus necator*. Sun et al. [132] constructed a dynamic model of *C. necator* DSM 545 to investigate polyhydroxybutyrate (PHB) synthesis during growth on glycerol. The dFBA framework allowed simulation of temporal changes in environmental conditions and revealed the metabolic adaptation of cells under nitrogen-limiting conditions. The model was calibrated using experimental measurements of extracellular fluxes and optimized by applying a flux minimization objective function, representing an energy-efficient metabolic behavior.

### 4.3. Assessment of Satellite Models Quality

We applied the MEMOTE [133] benchmarking framework to assess the quality of the curated satellite GSMs and their suitability for community reconstruction. The best-performing model was *i*CN1361 for *C. necator* H16, achieving a total score of 90% (Appendix A). However, only five out of the presented models achieved total scores above 70% (including *i*CN1361), indicating a limited number of high-quality curated models for satellites (Appendix A). A major issue was Stoichiometric Consistency test failures (0%) an important characteristic for model assessment. These failures indicate violations of mass conservation and the formation of thermodynamically infeasible loops. Additional problems with Mass Balance and Charge Balance were also identified in several models (Appendix A). This limited availability of reliable models poses a significant challenge.

On the other hand, automatically generated non-currated models often achieve high MEMOTE scores [133] and are widely used in workflows that derive community reconstructions from 16S or whole genome sequencing metagenomic data [134,135]. However their application is often constrained by reconstruction-related limitations: models may contain overrepresented reactions introduced during gap-filling or lack key metabolic pathways, which can substantially distort metabolite exchange profiles.

Nevertheless, all presented models can be used for integration into community models, but they require additional curation at the level of individual reconstructions in order to improve the predictive accuracy of the resulting community models.

## 5. Community Models of Methanotrophs and Satellites

Despite the availability of GSM models for both methanotrophs and their satellite organisms, the number of reconstructed genome-scale community models remains very limited. In this section, we summarize all published GSM-based community models involving methanotrophs and satellites, including one model constructed using dFBA. In addition, we introduce a subsection entitled “Other models,” which describes community GSM models not directly related to methanotroph–satellite systems and briefly covers other types of community-level modeling, such as co-occurrence-based approaches, that are not the main focus of this review but may still inform future reconstruction efforts.

### 5.1. Genome-Scale Community Models

At present, three published GSMs of microbial communities composed of methanotrophs and heterotrophs have been described. The first model represents a natural consortium reconstructed from metagenomic data. The other two models describe pairwise interactions between microorganisms under co-cultivation conditions and were created by integrating previously reconstructed individual GSM-models (see Section 4). The model of the natural community presented in the study by [34] was reconstructed using the CarveMe tool based on metagenomic data. To analyze interactions between strains, the SMETANA method was applied, enabling the assessment of cooperative and competitive relationships in a groundwater ecosystem. Among the identified methanotrophs were representatives of the order *Methylococcales*, as well as the genera *Methyloterricola* and *Methylocella*. However, it should be noted that the modeling results were limited to the analysis of sulfur and nitrogen compound fluxes, which did not allow for a comprehensive investigation of interactions among methanotrophs. This limitation highlights the importance of manual curation to improve the accuracy of predictions generated by automated tools such as CarveMe.

Unlike the study of a natural community, the work by [136] focused on a synthetic consortium consisting of the methanotroph *Methylomicrobium buryatense* 5GB1 (model *i*Mb5G(B1) [137]) and the cyanobacterium *Arthrospira platensis* NIES-39 [99]. A distinctive feature of this system is cultivation under anaerobic conditions, where *A. platensis* serves as the sole oxygen source for *M. buryatense*. An important aspect of this study was the prior acquisition of experimental data confirming interactions between the organisms via oxygen exchange, as well as precise measurements of growth rates, which were subsequently used to validate the model. Analysis of the reconstructed model not only confirmed the ability of *A. platensis* to produce oxygen and its consumption by *M. buryatense*, but also demonstrated good agreement between predicted and experimental growth rates. Detailed analysis revealed a complex network of metabolic interactions: *A. platensis* supplies *M. buryatense* with succinate and ammonium as carbon and nitrogen sources, respectively, while *M. buryatense* synthesizes amino acids required for the growth of the cyanobacterium.

The latest model [36] also represented a synthetic consortium of the methanotroph *Methylococcus capsulatus* Bath (model *i*McBath [138]) and *E. coli* strain K-12 substr. W3110 (model *i*EC1372_W3110 [106]). The study focused on detailed investigation of the interaction mechanisms between *M. capsulatus* and *E. coli* under different environmental limitations: specifically, oxygen and nitrate (as a single nitrogen source) limitations. Two variants of *E. coli* were considered: the wild-type and a genetically modified version engineered for homoserine production. Analysis of the models revealed active exchange of amino acids and nitrogen compounds between the consortium members. In the system with the wild-type *E. coli*, *M. capsulatus* produced acetate, which served as the sole carbon source for *E. coli*. Interestingly, in the community with the homoserine-producing *E. coli*, acetate production by *M. capsulatus* was reduced or ceased entirely, and malate was instead secreted to support the growth of *E. coli*.

### 5.2. Genome-Scale Community Models with dFBA

Earlier in this chapter, we described a study by Badr et al. [136] that used community modeling to analyze a synthetic microbial consortium composed of the methanotroph *Methylomicrobium buryatense* and the cyanobacterium *Arthrospira platensis.* In subsequent study [35], the consortium also analyzed using dynamic flux balance analysis (dFBA) to explore interspecies metabolic interactions during the co-utilization of biogas components. The primary biotechnological objective was to enhance methane and carbon dioxide conversion efficiency through the optimization of metabolic coupling between the two strains. To this end, the authors evaluated three genome-scale modeling approaches: the steady-state framework SteadyCom, dFBA using the DFBA Lab platform, and a novel hybrid approach, DynamiCom, developed by the authors. While dFBA allowed simulation of time-resolved growth under changing environmental conditions, it lacked the capacity to accurately capture interspecies interactions due to the absence of a shared metabolic milieu. This limitation motivated the development of DynamiCom, which integrates the temporal resolution of dFBA with interspecies metabolic coupling, enabling a more realistic representation of community dynamics and metabolic exchange.

### 5.3. Other Community Models

In the study by Islam et al. (2020) [139], the authors reconstructed a community composed of two functionally important methanotroph strains from Lake Washington. We did not include this model in the Section 5.1 because it described interaction between methanotrophs, without any heterotrophic satellites, but we think that the model can be extended in further studies.

The authors selected *Methylobacter tundripaludum* 21/22 and *Methylomonas* sp. LW13 because of their contributions to bioremediation, their ability to produce valuable metabolites, and the presence of annotated genomes.

The community model showed that *Methylobacter* dominates under nutrient-limited conditions, whereas *Methylomonas* prevails when nutrients are abundant. Flux balance analysis revealed that *Methylomonas* redirects a substantial fraction of carbon dioxide back into central metabolism via pyruvate decarboxylase. It was also noted that most carbon assimilation in both *Methylobacter* and *Methylomonas* occurs through the Entner–Doudoroff pathway.

Although *Methylobacter* is capable of assimilating methane more efficiently than *Methylomonas*, in co-culture it primarily utilizes formaldehyde produced by *Methylomonas*, which may be advantageous under oxygen-limited conditions by bypassing the methane oxidation reaction. It is important to note that the modeling results contradict experimental co-culture data reported by Yu et al. (2016) [140], although the community analyzed in this study included more than two species. Thus, the predictions require experimental validation for co-cultivation of *Methylobacter* and *Methylomonas*. Despite the discrepancy with empirical observations, the results provide valuable insights into interactions within microbial communities and the mechanisms of methane assimilation in aquatic ecosystems.

For methanotrophic microbial communities, there exists a co-occurrence network model of a community reconstructed by a research group [29] based on publicly available data. This analysis highlighted the central role of methanotrophs in community food web structure and revealed habitat-dependent differences in the composition of associated non-methanotrophic taxa. In particular, methanotrophs belonging to *Gammaproteobacteria* and *Alphaproteobacteria* were linked to distinct groups of heterotrophs, with stable associations suggested, for example, between *Methylosinus* (*Alphaproteobacteria*) and members of the orders *Rhizobiales* and *Rhodospirillales*, whereas *Gammaproteobacterial* methanotrophs were more frequently associated with *Pseudomonadales*. Across all investigated environments, methylotrophic genera such as *Methylotenera*, *Methylobacterium*, *Methylobacillus* and *Methylohalomonas* were consistently detected as co-occurring partners.

The co-occurrence network analysis identified both positive and negative correlations among community members but did not allow the underlying interaction mechanisms to be resolved. Nevertheless, such correlation-based approaches can be combined with GSM modeling or agent-based modeling in multi-scale frameworks, thereby increasing predictive power, as has been implemented, for example, in the BacArena platform [141].

## 6. Conclusions

Methanotrophic communities represent metabolically complex systems in which methane oxidation is tightly coupled to the activity of heterotrophic satellites. These partners not only remove inhibitory intermediates and provide essential cofactors but also critically shape the ecological stability and biotechnological potential of methanotroph-based systems. Despite the growing body of data on both natural and synthetic consortia, the mechanistic understanding of these interactions remains fragmented.

In this work, we reviewed the range of approaches used to study microbial communities, with particular emphasis on methanotrophic consortia. Special attention was given to methods of systems biology, where GSM modeling has emerged as a powerful framework for dissecting the internal interactions that support the stability of complex consortia. We systematically summarized available information on identified and potential satellites of aerobic methanotrophs and evaluated the current landscape of mathematical modeling applied to these organisms and their communities. The analysis reveals a clear gap between the number of experimentally characterized satellites and the availability of curated GSM models for them. In total, 23 models have been reported in the literature, but they cover only eight species. Four of these models correspond to closely related strains rather than the experimentally identified satellite. We systematically assessed the quality of the available models using the MEMOTE benchmarking framework [133] and only one model achieved Total Score quality above 90%. Despite the presence of single-species GSMs, community-level reconstructions for methanotroph–heterotroph consortia remain limited. To date, only five models involving methanotrophs have been reported: one co-occurrence network and four GSM-based reconstructions, of which only a single example incorporates dynamic flux balance analysis. These models provide only a partial representation of community-level metabolism and interspecies interactions in methanotroph–heterotroph consortia. To support future community-level modeling, we compiled the available GSM models of known methanotroph satellites and developed an open-access repository (https://gitlab.sirius-web.org/RSF/rsf_bioremediation, accessed on 14 December 2025) that aggregates the models and associated benchmarking outputs.

Future efforts will benefit from expanding community-level GSM modeling, including dFBA where appropriate, and from experiments designed to test model predictions. This is essential for moving from descriptive observations to mechanistic and predictive understanding of methanotrophic consortia.

## Figures and Tables

**Figure 1 microorganisms-14-00003-f001:**
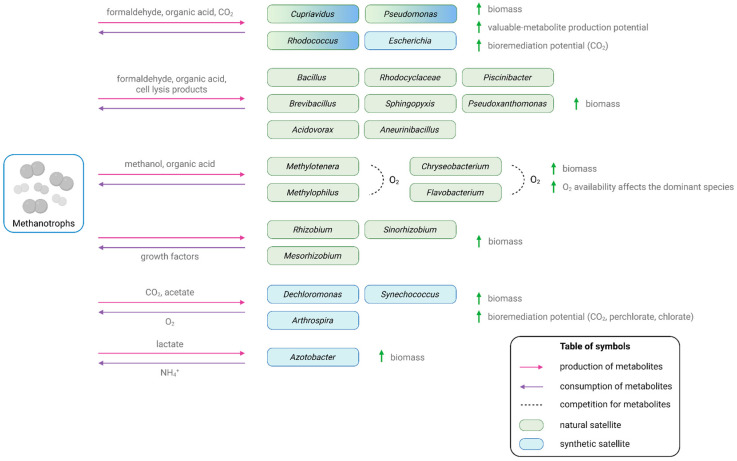
Functional roles of natural and synthetic methanotroph’s satellites in communities (created with BioRender.com).

**Table 1 microorganisms-14-00003-t001:** Comparison of systems biology approaches to the analysis of metabolic networks of microorganisms and interactions in communities.

Approach	A Mechanistic Description of Processes at the Organism Level	Application to M+H * Communities
Data-driven		
Co-occurrence network modeling	−	+
Time-series correlation	−	−
Causal inference models	−	−
Regression models	−	−
Mechanistic		
Ecological models		
Lotka–Volterra model	+/− (partly)	−
MacArthur’s model	+/− (partly)	−
Monod equation	+/− (partly)	+
Cell-level models		
Agent-based modeling	+	−
GSM-based models	+	+
Integrative and multiscale modeling approaches	+	−

* M+H—methanotrophs and heterotrophs.

**Table 2 microorganisms-14-00003-t002:** Identified Natural Satellites of Methanotrophs (Identified to the Species Level).

Methanotrophs	Satellites	Article
*Methylococcus capsulatus* Bath	*Cupriavidus necator*	[6]
	*Cupriavidus gilardii*	
	*Cupriavidus paucula*	
	*Brevibacillus agri*	
	*Brevibacillus formosus*	
	*Brevibacillus reuszeri*	
	*Brevibacillus choshiensis*	
	*Brevibacillus parabrevis*	
	*Brevibacillus brevis*	
	*Brevibacillus centrosporus*	
	*Brevibacillus borstelensis*	
	*Bacillus aneurinolyticus*	
	*Bacillus migulanus*	
	*Bacillus acidovorans*	
	*Bacillus thermoaerophilu*	
	*Bacillus laterosporus*	
	*Aneurinibacillus migulanus*	
	*Aneurinibacillus aneurinolyticus*	
*Methylococcus* sp. Concept-8	*Ochrobactrum intermedium* C13	[7]
	*Brevundimonas mediterranea* N7	
	*Cupriavidus* sp. S-6	
	*Ralstonia* sp. MSB2004	
	*Cupriavidus gilardii* CR3	
	*Chryseobacterium bernardetii* H4638	
	*Brevibacillus brevis* DZBY05	
	*Brevibacillus brevis* DZBY10	
	*Brevibacillus fluminis* CJ71	
*Methylomonas* sp. M5	*Cupriavidus taiwanensis* LMG 19424	[38]
*Methylocystis* sp. NLS7	*Pseudomonas chlororaphis*	[39]
*Methylosarcina fibrata* DSM 13736	*Staphylococcus aureus* R-23700	[40]
*Methylovulum miyakonense* HT12	*Rhizobium* sp. Rb122	[8]
*Methylosarcina*	*Chryseobacterium* sp. JT03	[41]
*Methylobacter* *Methylomonas*	*Methylophilus methylotrophus* Q8	[42]
	*Methylotenera mobilis* JLW8	
	*Methylotenera* sp. G11	
*Methylobacter tundripaludum*	*Methylotenera mobilis* JLW8	[43]
	*Methylotenera mobilis* 13	
*Methylosarcina*	*Methylophilus*	[43]
*Methylomonas* sp. *strain LW13*	*Methylophilus methylotrophus* Q8	[44]
	*Acidovorax* sp. 30s	
	*Flavobacterium* sp. 81	

**Table 3 microorganisms-14-00003-t003:** Experimentally validated potential methanotroph’s satellites.

Methanotrophs	Satellites	Article
*Methylococcus capsulatus* Bath	*Dechloromonas agitata* CKB	[64]
	*Escherichia coli* SBA01	[65]
*Methylotuvimicrobium buryatense* 5GB1C	*Azotobacter vinelandii* M5I3	[66]
*Methylomicrobium buryatense* 5GB1	*Arthrospira platensis* NIES-39	[67]
*Methylocystis hirsuta* CSC1	*Rhodococcus opacus* DSM 43205	[68]
	*Pseudomonas putida* KT2440	
*Methylocystis parvus* OBBP	*Rhodococcus opacus* DSM 43205	
	*Pseudomonas putida* KT2440	
*Methylomicrobium alcaliphilum* 20z	*Synechococcus* PCC 7002	[69]
*Methylocystis* sp. OK1	*Escherichia coli* BL21 (DE3)	[70]
*Methylomonas* spp.	*Rhizobium radiobacter* LMG 287	[71]
	*Ochrobactrum anthropi* LMG 2134	
	*Pseudomonas putida* LMG 24210	
	*Escherichia coli* LMG 2092T	
*Methylocystis parvus* OBBP	*Pseudomonas mandelii* JR-1	[72]
*Methylobacter luteus* 53v	*Bacillus pumilus* YXY-10	
	*Bacillus simplex* DUCC3713	
	*Exiguobacterium undae* B111	
	*Stenotrophomonas maltophilia* ATCC 13637	

**Table 4 microorganisms-14-00003-t004:** All Published GSM-models for methanotroph’s satellites.

Organism	Model Information				Article
	Model ID	Genes	Reactions	Metabolites	
*A. platensis* NIES-39	NIES-39	620	746	673	[99]
*A. vinelandii* DJ *	*i*DT1278	1278	2469	2003	[100]
	*i*AA1300	1300	2289	1958	[101]
*C. necator* H16 *	RehMBEL1391	1256	1391	1171	[102]
	RehMBEL1391—updated	1345	1538	1172	[103]
	*i*CN1361	1361	1292	1263	[104]
*E. coli* BL21(DE3)	*i*B21_1397	1397	2733	1943	[105]
	*i*ECD_1391	1391	2731	1943	
	*i*EC1356_Bl21DE3	1356	2740	1918	[106]
*P. putida* KT2440	*i*JN746	746	950	911	[107]
	*i*JP815	815	877	888	[108]
	*i*JP962	949	1066	980	[109]
	PpuMBEL1071	900	1071	1044	[110]
	PpuQY1140	1140	1171	1104	[111]
	*i*JN1462	1462	2929	2155	[112]
*R. opacus* PD630 *	*i*GR1773	1773	3025	1956	[113]
*S. aureus* USA300 str. JE2 *	*i*SB619	619	743	655	[114]
	*i*MH551	551	860	801	[115]
	*i*YS854	886	1455	1335	[116]
*Synechocystis* sp. PCC 7002	*i*Syp611	611	552	542	[117]
	*i*Syp708	708	646	581	[118]
	*i*Syp728	728	742	696	[119]
	*i*Syp821	821	792	777	[120]

* models of satellites assigned to the models of closely related strains group.

## Data Availability

The collection of satellite microorganism models, including GSM models, along with their MEMOTE reports, is available in the GitLab repository: https://gitlab.sirius-web.org/RSF/rsf_bioremediation (accessed on 13 December 2025).

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
