# Peer review of "From Single Organisms to Communities: Modeling Methanotrophs and Their Satellites"

_microorganisms, 2025, doi:10.3390/microorganisms14010003_

Round 1

Reviewer 1 Report

Comments and Suggestions for Authors

This review presenters an outlook at complex relationships between methanotrophs and non-methanotrophs and focuses on (computational) modeling approaches to dissecting such relationships, specifically pointing to the difficulties in applying modeling to the complex communities. In this sense this review is important and timely.  While heterotrophs are often juxtaposed with methanotrophs, technically, methanotrophs are heterotrophs. So perhaps you could use non-methanotrophs? While a large database is compiled of the observed methanotrophs satellites, I missed some frequently identified satellites such as Methylophilaceae (these are only briefly mentioned), Favobacteria or Burkholderiales. Their relationships with methanotrophs have not been investigated at the level of modeling, to my knowledge. However literature exists suggesting exchange of methanol and other simple organics. These organisms are good candidates for employment in synthetic communities, on the one hand, and are important in cycling methane across ecological niches. Perhaps you could mention them in your catalog. Please carefully check your references. For example Ho et al 2014 is cited as 55 and 83. Overall, I think bringing into light the necessity of combining different computational approaches followed by experimental verification is important not only for methanotroph communities but for other complex microbial  metabolisms.

Author Response

Response to reviewers

First of all, we would like to thank the reviewers for their critical comments and helpful suggestions. Based on these comments and suggestions, we have made careful modifications to the original manuscript. The reviewer’s comments are shown in black, followed by our responses in purple. The modifications made to the manuscript following the comments are marked as changes in editor mode.

Reviewer #1

Q:This review presenters an outlook at complex relationships between methanotrophs and non-methanotrophs and focuses on (computational) modeling approaches to dissecting such relationships, specifically pointing to the difficulties in applying modeling to the complex communities. In this sense this review is important and timely. While heterotrophs are often juxtaposed with methanotrophs, technically, methanotrophs are heterotrophs. So perhaps you could use non-methanotrophs? While a large database is compiled of the observed methanotrophs satellites, I missed some frequently identified satellites such as Methylophilaceae (these are only briefly mentioned), Favobacteria or Burkholderiales. Their relationships with methanotrophs have not been investigated at the level of modeling, to my knowledge. However literature exists suggesting exchange of methanol and other simple organics. These organisms are good candidates for employment in synthetic communities, on the one hand, and are important in cycling methane across ecological niches. Perhaps you could mention them in your catalog.

R: Thank you very much for your positive assessment of the review and for these thoughtful comments.We fully agree that, in a strict biochemical sense, methanotrophs are indeed heterotrophs. However, in ecological and microbial community studies, the term heterotrophs is frequently used in a functional sense to describe non-methanotrophic microorganisms that coexist with methanotrophs and rely on methanotroph-derived organic intermediates. To avoid ambiguity while maintaining this established ecological terminology, we have clarified the meaning of the term heterotrophs directly in the sentence where it first appears in the Introduction. The revised sentence now reads: “Accumulated data indicate that stable mixed cultures - comprising both methanotrophic and heterotrophic microorganisms, where heterotrophs refer to non-methanotrophic members that coexist with methanotrophs and utilize their metabolic by-products - are formed during long-term continuous cultivation of methanotrophs on natural gas under non-sterile conditions.”

This addition ensures that the intended ecological meaning of heterotrophs is explicit, while preserving the conventional terminology widely used in methanotroph community research.

We thank the reviewer for highlighting the insufficient description of key satellites of methanotrophs. In response, we expanded Section 3.1.1 “Satellites that promote a methanotroph’s growth by consuming metabolic by-products”, providing a more detailed overview of the permanent observed family Methylophilaceae and adding information on the genus Acidovorax, a member of the order Burkholderiales. We also note that our catalog already included commonly reported representatives from the class Flavobacteria – the genera Flavobacterium and Chryseobacterium – as well as the genus Cupriavidus from Burkholderiales.

Q:Please carefully check your references. For example Ho et al 2014 is cited as 55 and 83. R:Thank you for your comment. We have checked all the references and corrected the inconsistencies.

Q:Overall, I think bringing into light the necessity of combining different computational approaches followed by experimental verification is important not only for methanotroph communities but for other complex microbial metabolisms.

R:Thank you for your comment. We agree that our original wording could be interpreted as referring only to methanotrophic communities. We have clarified this point and adjusted the phrasing in both Abstract and Conclusion.

References

Ceballos Rodriguez-Conde, F., Zhu, S., and Dikicioglu, D. (2025). Harnessing microbial division of labor for biomanufacturing: a review of laboratory and formal modeling approaches. Critical Reviews in Biotechnology 45, 1249–1267. doi: 10.1080/07388551.2025.2455607

Crumbley, A. M., Garg, S., Pan, J. L., and Gonzalez, R. (2024). A synthetic co-culture for bioproduction of ammonia from methane and air. Journal of Industrial Microbiology and Biotechnology 51, kuae044. doi: 10.1093/jimb/kuae044

Gong, T., Liu, R., Che, Y., Xu, X., Zhao, F., Yu, H., et al. (2016a). Engineering Pseudomonas putida KT 2440 for simultaneous degradation of carbofuran and chlorpyrifos. Microbial Biotechnology 9, 792–800. doi: 10.1111/1751-7915.12381

Gong, T., Liu, R., Zuo, Z., Che, Y., Yu, H., Song, C., et al. (2016b). Metabolic Engineering of Pseudomonas putida KT2440 for Complete Mineralization of Methyl Parathion and

γ-Hexachlorocyclohexane. ACS Synth. Biol. 5, 434–442. doi: 10.1021/acssynbio.6b00025 Gong, T., Xu, X., Dang, Y., Kong, A., Wu, Y., Liang, P., et al. (2018). An engineered Pseudomonas putida can simultaneously degrade organophosphates, pyrethroids and carbamates. Science of The Total Environment 628–629, 1258–1265. doi: 10.1016/j.scitotenv.2018.02.143

Oña, L., Shreekar, S. K., and Kost, C. (2025). Disentangling microbial interaction networks. Trends in Microbiology 33, 619–634. doi: 10.1016/j.tim.2025.01.013

Plaggenborg, R., Overhage, J., Steinbüchel, A., and Priefert, H. (2003). Functional analyses of genes involved in the metabolism of ferulic acid in Pseudomonas putida KT2440. Appl Microbiol Biotechnol 61, 528–535. doi: 10.1007/s00253-003-1260-4

Qian, Y., Lan, F., and Venturelli, O. S. (2021). Towards a deeper understanding of microbial communities: integrating experimental data with dynamic models. Current Opinion in Microbiology 62, 84–92. doi: 10.1016/j.mib.2021.05.003

Santillan, E., Neshat, S. A., and Wuertz, S. (2025). Disturbance and stability dynamics in microbial communities for environmental biotechnology applications. Current Opinion in Biotechnology 93, 103304. doi: 10.1016/j.copbio.2025.103304

Van Den Berg, N. I., Machado, D., Santos, S., Rocha, I., Chacón, J., Harcombe, W., et al. (2022). Ecological modelling approaches for predicting emergent properties in microbial communities. Nat Ecol Evol 6, 855–865. doi: 10.1038/s41559-022-01746-7

Reviewer 2 Report

Comments and Suggestions for Authors

Overall comments
The manuscript provides a timely overview of metabolic modeling efforts related to methanotrophs and their associated satellite organisms. The topic is relevant, and the structure is generally clear. However, several issues reduce the coherence and usefulness of the review. The connection between methodological approaches (Table 1) and the actual reconstructed models (Table 4) is not clearly established, making it difficult to understand how specific modeling frameworks have been applied in practice. Some sections contain sudden shifts in focus, for example, the mention of Azotobacter vinelandii M5I3 appears without context, and the distinction between individual satellite models and community modeling could be articulated more clearly. In addition, co-occurrence network analysis is presented alongside mechanistic modeling approaches, even though it is only a correlation-based method and not directly comparable to GSM- or FBA-based frameworks. Clarifying these conceptual boundaries and strengthening the link between models and methods would significantly improve the manuscript.

Figure 1 presents a comprehensive framework of experimental, GSM-based, and dynamic/statistical modeling approaches. However, the manuscript does not fully cover several categories highlighted in the figure, particularly ecological models, agent-based modeling, dynamic modeling, and regression-based approaches, which makes the figure feel disconnected from the main text. Expanding these sections or adjusting the figure for consistency would improve the coherence of the review.

Section 2 is presented in a very general and high-level manner, and several parts read more like generic AI-generated text than a focused scientific analysis. The statements are often broad, repetitive, and lack concrete examples or mechanistic explanation. In a few places, the wording becomes circular, for example, describing systems biology approaches as ‘helping to understand complex systems’ without specifying how, or which methods are relevant to methanotroph–satellite interactions. As a result, the section does not provide substantive insight into the actual advantages, limitations, or applications of these approaches. Table 1 is similarly too broad, listing categories without discussing their principles, assumptions, or relevance to the specific microbial systems reviewed. The authors are encouraged to revise this section with more critical depth, method-specific detail, and contextually grounded examples.

Specific comments: 

Line 20: What is GSMs? Please write the full name when first meet to readers.
Line 34: MMO, same as the above comments.
Line 55, ”is/are” is missing?
Line 65, same as the first two comments.
Line 99, reference is missing.
Line 129, why “more difficult to interpret”? please add more details.
Line 130- 135, very repetitive sentences. 
Line 161, unclear sentence, stimulate who’s growth?
Line 241, please define what is ”biosafety strain” 
Table 4. The manuscript would benefit from indicating, in Table 4, the specific modeling approaches used for each model, as the current version does not connect the approaches in Table 1 with the models listed.

Line 426-427, The reference to A. vinelandii M5I3 appears abrupt. The manuscript should explain why this strain is discussed, how it relates to methanotrophic satellites, and why its GSM availability is important in this context.
Line 500, The conceptual distinction between Section 4 (individual metabolic models of satellites) and Section 5 (community models integrating methanotrophs and satellites) is valid. However, the manuscript does not clearly articulate this boundary, and the examples in both sections partly overlap. Adding explicit clarification of what differentiates individual GSMs from community modeling frameworks would improve clarity and logical flow.
Line 504: Co-occurrence network analysis is correlation-based and does not provide mechanistic insight into metabolic interactions. In contrast, GSM- and FBA-based approaches are stoichiometrically grounded mechanistic models capable of predicting metabolic fluxes, nutrient exchange, and community behavior. The manuscript should clearly distinguish between statistical association methods and mechanistic metabolic modeling

Comments on the Quality of English Language

The English is generally understandable, but several sections would benefit from substantial revision for clarity and coherence. Many paragraphs lack logical transitions, resulting in abrupt shifts between concepts. In addition, some sentences read overly generic and formulaic, resembling AI-generated text, with repetitive phrasing and limited substantive content. Strengthening the logical flow, reducing vague or circular statements, and providing more precise, context-specific wording would significantly improve the readability and scientific clarity of the manuscript.

Author Response

Response to reviewers

First of all, we would like to thank the reviewers for their critical comments and helpful suggestions. Based on these comments and suggestions, we have made careful modifications to the original manuscript. The reviewer’s comments are shown in black, followed by our responses in purple. The modifications made to the manuscript following the comments are marked as changes in editor mode.

Reviewer#2 Overall comments

The manuscript provides a timely overview of metabolic modeling efforts related to

methanotrophs and their associated satellite organisms. The topic is relevant, and the structure is generally clear. However, several issues reduce the coherence and usefulness of the review.

Q:The connection between methodological approaches (Table 1) and the actual reconstructed models (Table 4) is not clearly established, making it difficult to understand how specific modeling frameworks have been applied in practice.

R: Thank you for this helpful comment. We agree that, in the previous version, the relationship between the methodological overview and the list of reconstructed models was not sufficiently clear. Table 1 summarizes the main classes of approaches used for the reconstruction and analysis of community-level models, whereas Table 4 lists available genome-scale metabolic models for individual methanotroph satellites. These organism-level GSMs are intended to serve as building blocks for community models that can be constructed using the frameworks described in Chapter 2 and summarized in Table 1.

To clarify this connection for the reader, we have now revised the caption and introductory sentence of Table 4 to explicitly state that these satellite models are meant for subsequent use in community modelling, and added a short bridging paragraph at the end of Chapter 2 / beginning of Chapter 4 explaining how the approaches in Table 1 can be applied to build community models based on the GSMs listed in Table 4. We have also slightly adjusted the wording in Table 1 to emphasize more clearly that it focuses on methods for community-level model reconstruction.

Q:Some sections contain sudden shifts in focus, for example, the mention of Azotobacter vinelandii M5I3 appears without context, and the distinction between individual satellite models and community modeling could be articulated more clearly.

R:Thank you for this comment. We would like to clarify that Azotobacter vinelandii M5I3 is introduced and discussed in detail in Chapter 3, where it is described as a satellite strain used in synthetic co-culture with Methylomicrobium buryaten (Crumbley et al., 2024) in Section 3.2.1 (“Synthetic methanotrophic communities for the production of metabolites”). We agree, however, that the transition in Chapter 4 was not sufficiently smooth in the previous version. To make the connection explicit, we have revised the opening sentence of the relevant paragraph in Chapter 4 to: “In Chapter 3, the Azotobacter vinelandii M5I3 strain was described as a methanotroph satellite, but there are currently no strain-specific GSM models available for this organism.” During revision, we also identified a typo in Table 2,

where the strain was incorrectly labeled as “A. vinelandii M5I”; this has now been corrected to “A. vinelandii M5I3.”

Regarding the distinction between individual satellite models and community modelling, we have expanded Chapter 4 to clarify this more clearly. We now explicitly state that

genome-scale metabolic models of individual satellite organisms are treated as building blocks that can be combined in subsequent community-level reconstructions, in contrast to community models that represent multiple species and their interactions within a single integrated framework. In addition, we have added a brief description of the available individual satellite models and their suitability for community-level model reconstruction at the beginning of Chapter 4 to improve the conceptual link between Chapters 2, 4 and 5.

Q:In addition, co-occurrence network analysis is presented alongside mechanistic modeling approaches, even though it is only a correlation-based method and not directly comparable to GSM- or FBA-based frameworks. Clarifying these conceptual boundaries and strengthening the link between models and methods would significantly improve the manuscript.

R:Thank you for your comment. We agree that co-occurrence networks do not reflect mechanistic interactions, unlike GSM models. In the revised version, we now state this distinction more clearly in Chapter 2, where co-occurrence networks are discussed together with other statistical approaches and explicitly contrasted with mechanistic modelling frameworks. Our main focus in the review remains on GSM-based community models of methanotrophs. We chose to retain the co-occurrence network study because it is, to our knowledge, the only published community-level analysis of methanotrophic systems using this type of approach. Initially, this model was assigned to a dedicated subsection, but we agree that the focus should be shifted toward GSM-based approaches. Therefore, we revised Chapter 5 and moved the co-occurrence network model to the subsection Other community models.

Q:Figure 1 presents a comprehensive framework of experimental, GSM-based, and dynamic/statistical modeling approaches. However, the manuscript does not fully cover several categories highlighted in the figure, particularly ecological models, agent-based modeling, dynamic modeling, and regression-based approaches, which makes the figure feel disconnected from the main text. Expanding these sections or adjusting the figure for consistency would improve the coherence of the review.

R: Thank you for your comment. We agree that some of the categories shown there (ecological, agent-based, dynamic and regression-based approaches) were only briefly discussed in the text and that this could make the graphical abstract feel disconnected from the main focus of the review. Since our primary emphasis is on GSM-based modelling of microbial communities, we have now revised the graphical abstract to better reflect the actual scope of the manuscript, simplifying the framework and removing elements that are not treated in sufficient detail in the main text.

Q:Section 2 is presented in a very general and high-level manner, and several parts read more like generic AI-generated text than a focused scientific analysis. The statements are often broad, repetitive, and lack concrete examples or mechanistic explanation. In a few places, the wording becomes circular, for example, describing systems biology approaches as ‘helping to understand complex systems’ without specifying how, or which methods are relevant to methanotroph–satellite interactions. As a result, the section does not provide

substantive insight into the actual advantages, limitations, or applications of these approaches. Table 1 is similarly too broad, listing categories without discussing their principles, assumptions, or relevance to the specific microbial systems reviewed. The authors are encouraged to revise this section with more critical depth, method-specific detail, and contextually grounded examples.

R: Thank you for your recommendation. We agree that Chapter 2 is presented in a general and high-level manner. This was intentional, due to the large number of recent publications – including several from this year-that already provide detailed overviews of approaches for modeling microbial communities at various levels, along with their respective advantages and limitations. Although the initial draft included an in-depth description of all these methods, we ultimately decided that such detail would largely duplicate existing reviews (Qian et al., 2021; Van Den Berg et al., 2022; Ceballos Rodriguez-Conde et al., 2025; Oña et al., 2025; Santillan et al., 2025). Therefore, we opted to give only a concise overview of the methods, supplemented by references to more comprehensive reviews. As you correctly pointed out in your comment, we inadvertently omitted the links to these reviews - an oversight for which we sincerely apologize. These references have now been added to the text.

We significantly revised Chapter 2, where we briefly characterize the key reviews that cover the general methodology, and avoid circular wording such as “systems biology approaches help to understand complex systems” without further explanation. For each class of methods discussed (statistical/correlation-based, ecological, agent-based and GSM-based modelling), we now provide more method-specific detail on their main assumptions, typical applications, and the type of insight they offer. Where possible, we illustrate these points with examples that are directly relevant to methanotroph–satellite communities or how it can be implemented in their context.

Table 1 has also been revised in line with these changes. Rather than serving as a purely generic list of categories, it now summarizes the main approaches discussed in Chapter 2 and contrasts them along two key aspects: whether they provide a mechanistic description at the organism level and whether they have been applied to methanotroph–heterotroph communities. We clarify this scope in the table caption and in the surrounding text, so that readers can more easily see which modelling frameworks are currently used in this specific context and where important gaps remain.

Specific comments:

Q:Line 20: What is GSMs? Please write the full name when first meet to readers.

R:Thank you for your comment. Replaced to the full name of the term in the main text.

Q:Line 34: MMO, same as the above comments.

R:Thank you for your comment. The full name was added in the main text.

Q:Line 55, ”is/are” is missing?

R:Thank you for your comment. We have corrected the sentence to “However, the structure of methanotrophic communities, as well as the mechanisms of interactions between main bacterial strain and its satellites, remains insufficiently understood, primarily due to methodological and analytical constraints”.

Q:Line 65, same as the first two comments.

R:Thank you for your comment. Fixed in the text.

Q:Line 99, reference is missing.

R:Thank you for your comment. Fixed in the text.

Q:Line 129, why “more difficult to interpret”? please add more details.

R: Thank you for your comment. The Chapter 2 text was significantly revised.

Q:Line 130- 135, very repetitive sentences.

R: Thank you for your comment. The Chapter 2 text was significantly revised.

Q:Line 161, unclear sentence, stimulate who’s growth?

R:Thank you for your comment. We have clarified the sentence in the manuscript to specify that satellites stimulate the growth of methanotrophs.

Q:Line 241, please define what is ”biosafety strain”

R: Thank you for your comment. In this context, the term “biosafety strain” refers to a non-pathogenic bacterium classified as biosafety level 1 that is widely used in biotechnological applications. For Pseudomonas putida KT2440, this terminology is

commonly used in the literature (Plaggenborg et al., 2003; Gong et al., 2016b, 2016a, 2018). We agree that the term may be unclear for some readers, and we have therefore replaced it in the manuscript with a more explicit description: “Pseudomonas putida KT2440 is classified as a non-pathogenic strain (biosafety level 1) that is widely used in biotechnological processes.”

Q:Table 4. The manuscript would benefit from indicating, in Table 4, the specific modeling approaches used for each model, as the current version does not connect the approaches in Table 1 with the models listed.

R: Thank you for this helpful comment. We agree that, in the previous version, the relationship between the methodological overview and the list of reconstructed models was not sufficiently clear. Table 1 summarizes the main classes of approaches used for the reconstruction and analysis of community-level models, whereas Table 4 lists available genome-scale metabolic models for individual methanotroph satellites. These organism-level GSMs are intended to serve as building blocks for community models that can be constructed using the frameworks described in Chapter 2 and summarized in Table 1.

To clarify this connection for the reader, we have now revised the caption and introductory sentence of Table 4 to explicitly state that these satellite models are meant for subsequent use in community modelling, and added a short bridging paragraph at the end of Chapter 2 / beginning of Chapter 4 explaining how the approaches in Table 1 can be applied to build community models based on the GSMs listed in Table 4. We have also slightly adjusted the wording in Table 1 to emphasize more clearly that it focuses on methods for community-level model reconstruction.

Q:Line 426-427, The reference to A. vinelandii M5I3 appears abrupt. The manuscript should explain why this strain is discussed, how it relates to methanotrophic satellites, and why its GSM availability is important in this context.

R: Thank you for this comment. We would like to clarify that Azotobacter vinelandii M5I3 is introduced and discussed in detail in Chapter 3, where it is described as a satellite strain used in synthetic co-culture with Methylomicrobium buryatense 5GB1 (Crumbley et al., 2024) in Section 3.2.1 (“Synthetic methanotrophic communities for the production of metabolites”). We agree, however, that the transition in Chapter 4 was not sufficiently smooth in the previous version. To make the connection explicit, we have revised the opening sentence of the relevant paragraph in Chapter 4 to: “In Chapter 3, the Azotobacter vinelandii M5I3 strain was described as a methanotroph satellite, but there are currently no strain-specific GSM models available for this organism.” During revision, we also identified a typo in Table 2, where the strain was incorrectly labeled as “A. vinelandii M5I”; this has now been corrected to “A. vinelandii M5I3.”

Q:Line 500, The conceptual distinction between Section 4 (individual metabolic models of satellites) and Section 5 (community models integrating methanotrophs and satellites) is valid. However, the manuscript does not clearly articulate this boundary, and the examples in both sections partly overlap. Adding explicit clarification of what differentiates individual GSMs from community modeling frameworks would improve clarity and logical flow.

R: Thank you for this helpful comment. We have revised the text to make the distinction between Sections 4 and 5 more explicit. In Section 4 (“Mathematical models of methanotroph’s satellites”), we have rewritten the introductory paragraph and added a brief concluding statement to clarify that this section focuses on genome-scale metabolic models of individual satellite organisms and their use as building blocks for subsequent community reconstructions. In Section 5 (“Community models of methanotrophs and satellites”), we have revised the introduction to explicitly contrast these organism-level GSMs with community-level modelling frameworks, which represent multiple species and their interactions within a single model. We hope that these changes improve the clarity and logical flow between the two sections.

Q:Line 504: Co-occurrence network analysis is correlation-based and does not provide mechanistic insight into metabolic interactions. In contrast, GSM- and FBA-based approaches are stoichiometrically grounded mechanistic models capable of predicting metabolic fluxes, nutrient exchange, and community behavior. The manuscript should clearly distinguish between statistical association methods and mechanistic metabolic modeling

R: Thank you for your comment. We agree that co-occurrence networks do not reflect mechanistic interactions, unlike GSM models. In the revised version, we now state this distinction more clearly in Chapter 2, where co-occurrence networks are discussed together with other statistical approaches and explicitly contrasted with mechanistic modelling frameworks. Our main focus in the review remains on GSM-based community models of methanotrophs. We chose to retain the co-occurrence network study because it is, to our knowledge, the only published community-level analysis of methanotrophic systems using this type of approach. Initially, this model was assigned to a dedicated subsection, but we agree that the focus should be shifted toward GSM-based approaches. Therefore, we revised Chapter 5 and moved the co-occurrence network model to the subsection Other community models.

Round 2

Reviewer 1 Report

Comments and Suggestions for Authors

The Authors sufficiently addressed my suggestions and the revised versión is a nice work. 

Reviewer 2 Report

Comments and Suggestions for Authors

The revised version shows substantial improvement in content and organization. Compared to the previous version, the logical flow between sections is much clearer, and the addition of specific examples greatly enhances clarity and traceability, readers can now easily find original references to support key points. However, the introduction begins by describing methane oxidation enzymes, yet methanogens, despite being the primary biological source of methane and thus a key upstream component of methanotrophs’ substrate supply, are not discussed throughout the review. This makes the scope feel somewhat incomplete in the context of the broader methane cycle. A brief explanation of why methanogens were excluded from the modeling frameworks would help clarify the scope and strengthen the coherence of the review.

Lines 128–159: The division into “statistical and correlation methods” vs. “mathematical models” is functionally useful but conceptually inconsistent. Statistical models, such as regression and causal inference, are themselves mathematical. A clearer classification (e.g., data-driven vs. mechanistic) would improve precision. Additionally, Table 1 lists “statistical and correlation methods” alongside “ecological” and “cell-level” models, omitting the “mathematical model” category entirely, which adds to the confusion.

Line 321 onward: This section contains content that overlaps with earlier parts of the review. Please check for redundancy.

Lines 362–365: This sentence repeats earlier points. Also, the phrase “probably likely” is redundant—please revise.

Line 569: The full form of GSM is not necessary here, as it has already been defined.
